# Correlation between the Cycle Threshold Values in Detection of Severe Fever with Thrombocytopenia Syndrome Virus Using PowerChek^TM^ SFTSV Real-Time PCR Kit and Viral Load: Prognostic Implications

**DOI:** 10.3390/v16050700

**Published:** 2024-04-29

**Authors:** Misun Kim, Sang Taek Heo, Hee Cheol Kim, Myeong Jin Kang, Sora Kim, Keun Hwa Lee, Jeong Rae Yoo

**Affiliations:** 1Department of Internal Medicine, Jeju National University Hospital, Jeju 63241, Republic of Korea; drkms1016@gmail.com (M.K.); neosangtaek@naver.com (S.T.H.); hck950703@gmail.com (H.C.K.); kaej159@naver.com (M.J.K.); thfk130@hanmail.net (S.K.); 2Department of Internal Medicine, Jeju National University College of Medicine, Jeju 63241, Republic of Korea; 3Department of Microbiology, Hanyang University College of Medicine, Seoul 04763, Republic of Korea; yomust7@gmail.com

**Keywords:** severe fever with thrombocytopenia syndrome, *Banyangvirus*, tick-borne disease, cycle threshold

## Abstract

Background: This study aimed to analyze the correlation between the cycle threshold (Ct) values of severe fever with thrombocytopenia syndrome (SFTS) virus small (S) and middle (M) segments and the SFTS viral load, aiming to estimate the initial viral load and predict prognosis in the early clinical course. Method: A retrospective study was conducted with confirmed SFTS patients at Jeju National University Hospital (2016–2022). Patients were categorized into non-fatal and fatal groups. Results: This study included 49 patients with confirmed SFTS (non-fatal group, *n* = 42; fatal group, *n* = 7). A significant negative correlation (−0.783) was observed between the log SFTS viral load and Ct values (*p* < 0.001). This negative correlation was notably stronger in the fatal group (correlation coefficient −0.940) than in the non-fatal group (correlation coefficient −0.345). Conclusion: In this study, we established a correlation between SFTS viral load and Ct values for estimating the initial viral load and early predicting prognosis. These results are expected to offer valuable insights for SFTS patient treatment and prognosis prediction.

## 1. Introduction

Severe fever with thrombocytopenia syndrome (SFTS) is an emerging tick-borne disease caused by the SFTS virus (SFTSV) belonging to the genus *Banyangvirus*, family *Phenuiviridae*, and order *Bunyavirales* [1]. Despite being known for more than a decade, there was no specific treatments for patients with SFTS leading to a mortality rate of approximately 5–30% in East Asia [2,3]. Hence, the early identification of poor prognostic factors associated with fatal clinical outcome is of paramount importance in clinical practice. Previous studies reported that a high SFTS viral load in the blood in the acute clinical phase was associated with high mortality [4,5]. Hence, early measurement of the SFTS viral load is important for diagnosis and prognosis prediction. However, few studies have been conducted on laboratory findings or tools that can indirectly predict SFTS viral load, because quantitative analysis of the SFTS RNA viral load is difficult and time-consuming to test in a general laboratory.

The genome of SFTSV comprises three negative single-stranded RNA segments: large (L), medium (M), and short (S) [6]. The Jeju Research Institute of Public Health and Environment uses real-time reverse transcription–polymerase chain reaction to confirm and diagnose SFTS according to the guideline of the Korea Disease Control and Prevention Agency, i.e., by detecting the S and M segments of the SFTSV using PowerCheck^TM^ SFTSV Real-time PCR kit [7]. This method takes 4 h, and the results are reported on the same day or next day in the Jeju Island area. Determining if there is a correlation between the SFTS viral load and the cycle threshold (Ct) values of the S and M segments within a short period may enable the estimation of the SFTS viral load and rapid prediction of disease severity based on Ct values. This study aimed to analyze the relationship between Ct values and SFTS viral load of the SFTSV to facilitate more efficient disease management.

## 2. Materials and Methods

This retrospective observational study was conducted in Jeju National University Hospital, a single tertiary hospital in Jeju Island, Republic of Korea, between March 2016 and December 2022. Patients with confirmed SFTS were divided into non-fatal (survival) and fatal (death) groups. The demographic and clinical characteristics data were obtained from the electronic medical records and included the patient demographics, presence of initial symptoms (fever, chills, fatigue, headache, myalgia, dizziness, poor oral intake, nausea, vomiting, abdominal pain, diarrhea, hemoptysis, and dyspnea), laboratory parameters, past medical history, Charlson Comorbidity Index score (CCI), and multiple organ dysfunction score (MODS) during hospitalization and after 72 h of treatment [8,9]. The inclusion criteria are the following: patients whose blood samples were collected during the first visit to the hospital; patients whose Ct values of the S and M segments of the SFTSV RNA and SFTS viral load were assessed using a same day of serum sample. The Ct values of the S and M segment gene of the SFTSV RNA were evaluated using the SFTSV real-time RT-PCR Kit (PowerChek; Kogene Biotech, Seoul, Republic of Korea) at the Jeju Research Institute of Public Health and Environment (Powercheck S-Ct value and M-Ct value, hereinafter referred to S-segment and M-segment Ct value in the text). Analysis using the PowerChek kit was performed on the Bio-Rad CFX96 system (Bio-Rad Laboratories Inc., Hercules, CA, USA) in a total volume of 20 µL (15 µL of PCR mixture and 5 µL of template RNA), according to the manufacturer’s instructions. Samples with a Ct value equal to or less than 38 were reported as positive for SFTS viral-genome detection. The presence of SFTSV and its RNA copies were evaluated using RT-PCR in the Department of Microbiology and Immunology, Jeju National University College of Medicine, Jeju, Republic of Korea. Viral RNA was extracted from the first acute-phase serum of a confirmed SFTS patient using a QIAamp Viral RNA Mini kit (Qiagen Inc., Mainz, Germany). The extracted RNA was preserved in elution buffer at −70 °C. The RT-PCR of the partial S and M segments of SFTSV was performed. The RT-PCR mixture contained 8 µL of one-step RT-PCR premix, 7 µL of detection solution, and 5 µL of the RNA template (total volume of 20 µL). The following cycling conditions were used: 30 min at 45 °C, 10 min at 90 °C, 45 cycles of 15 s at 95 °C and 30 s at 48 °C. The products were sequenced using a BigDye Terminator Cycle Sequencing kit (Perkin Elmer Applied Bio-systems, Warrington, UK). Descriptive statistics were presented as frequencies and percentages for categorical variables and as means and standard deviations for continuous variables. We compared baseline demographics, clinical characteristics, and initial laboratory results among the study population using either Student’s *t*-test, the Mann–Whitney test, a Chi-squared test, or Fisher’s exact test. We also compared the significance according to the Ct value intervals using the Kruskal–Wallis test. Person’s correlation analysis was performed to evaluate the correlation of Ct value and SFTS viral load. By the rule of thumb for interpreting the size of a correlation coefficient, 0.0 to 0.3 (0.0 to −0.3) is negligible, 0.3 to 0.5 (−0.3 to −0.5) is low positive (negative), 0.5 to 0.7 (−0.5 to −0.7) is moderate positive (negative), 0.7 to 0.9 (−0.7 to −0.9) is high positive (negative), and 0.9 to 1.0 (−0.9 to −1.0) is very high positive (negative) correlation. All statistical analyses were performed using SPSS version 25.0 (IBM Corp., Armonk, NY, USA).

## 3. Results

Among the 67 patients with confirmed SFTS during the study period, 49 patients were included in the analysis, of whom 49 had Ct values for the S segment and 34 had Ct values for both S and M segments available for analysis. Of the 49 patients, 42 were in the non-fatal group and 7 were in the fatal group (Appendix A). The baseline clinical characteristics and laboratory findings are summarized in Table 1. The Charlson Comorbidity Index score was significantly higher in the fatal group than in the non-fatal group. The difference in the initial log SFTS viral load between the groups was also not statistically significant. There was no significant difference between the Ct values of the S and M segment. There was also no significant difference between the Ct values of the S and M segments in terms of the initial SFTS viral load or mortality rate. At 72 h post-diagnosis, the fatal group exhibited a significantly higher mean Multiple Organ Dysfunction Score than the non-fatal group. Despite the lack of statistical significance, a higher proportion of the fatal group underwent therapeutic plasma exchange (TPE) than the non-fatal group.

We observed a substantial negative correlation between the SFTS viral load and the Ct values of the S and M segments. As there was no statistically significant difference between the Ct values of the S and M segment, the statistical analysis was performed using S-segment, which has a relatively large sample size. In the correlation analysis of the SFTS viral load and Ct value of the S segments, the SFTS viral load and log SFTS viral load had correlation coefficients of −0.783 (*p* < 0.001) and −0.628 (*p* < 0.001), respectively (Figure 1A), which implies that as the Ct value decreases, the SFTS viral load increases. Importantly, this correlation was more pronounced in the fatal group with correlation coefficients of −0.940 (*p* = 0.002) for the SFTS viral load and −0.816 (*p* = 0.025) for the log SFTS viral load as opposed to −0.345 (*p* = 0.025) and −0.447 (*p* = 0.003), respectively, in the non-fatal group (Figure 1B,C).

To predict the SFTS viral load corresponding to the interval of the Ct value, we analyzed the mean viral load for different Ct value intervals. When the Ct values were divided into five intervals (<21, 21–25, 25–29, 29–33, and >33), the mean viral loads of the 21–25 and >29–33 intervals were 602,221 ± 1,013,744 (median, 273.587; interquartile range (IQR), 571–945) and 106,120 ± 204,463 (median, 11,514; IQR, 76–165), respectively, and the difference in the mean viral load between Ct values of 21–25 and 29–33 was statistically significant (*p* = 0.031). The number of fatal cases tended to increase in the >25–29 interval. Similarly, there was a statistically significant difference in the mean log viral load between Ct values of 21–25 and 29–33 (*p* = 0.031) (Figure 2).

## 4. Discussion

This study analyzed the relationship between the SFTS viral load and the Ct values of the S and M segments of the SFTSV, shedding light on the potential implications for disease management and prognosis prediction. To the best of our knowledge, this is the first study to statistically evaluate the correlation between the SFTS viral load and Ct value. Our analysis revealed a strong negative correlation between the Ct values of the S and M segments and the SFTS viral load, suggesting that as the Ct value decreases, the SFTS viral load increases. This correlation was more pronounced in the fatal group, highlighting the potential utility of Ct values in predicting disease severity. Examination of the mean viral load at different Ct intervals showed statistically significant differences in the mean viral load between specific Ct value intervals, particularly between Ct values of 21–25 and 29–33. These findings suggest that Ct values may serve as valuable indicators for estimating the SFTS viral load and predicting disease outcomes.

A recent study investigating the factors associated with mortality in patients with COVID-19 analyzed the relationship between the viral load and Ct values. In studies of SARS-CoV-2 viral load and Ct values, the SARS-CoV-2 Ct values provided by the QIAstat-Dx^®^ Respiratory SARS-CoV-2 panel could be used as a surrogate for viral load, given the linear correlation between Ct values and viral load, and individuals with higher SARS-CoV-2 viral load demonstrated lower Ct levels, which correlates with the high mortality in patients with COVID-19 [10,11]. In another COVID-19 study, COVID-19 patients with high viral load (Ct value <25) had increased intubation risk and higher mortality rates [12]. This finding suggests a potential correlation between the Ct value and viral load in viral disease. In studies examining the association between SFTS viral copies and mortality rates, fatal SFTS patients often exhibited higher viral loads than non-fatal patients [4,13]. However, a previous study has evaluated the factors associated with mortality in SFTS patients, and the results showed a significant correlation between the SFTS viral load and the Ct value in both survival and non-survival groups, which aligns with the results of our study [14]. We further divided Ct values into intervals to analyze the mean SFTS viral load within each Ct value interval, which allowed us to surmise the association of high mortality below the certain Ct value.

Although the SFTS viral load correlates with mortality, researchers have conducted numerous studies to explore factors other than the Ct value for predicting SFTS viral load. One study that investigated the relationship between viral load and serum cytokines or chemokines, such as interferons (IFN) α and γ, interleukin (IL) 10, monocyte chemoattractant protein 1, C-X-C motif chemokine ligand 8, and INF-γ–induced protein 10, found positive correlations [15]. In another study, the SFTS viral load correlated with the level of several peripheral cytokines, including IFN-γ, heat-shock protein 70, and IL-15 [16]. Furthermore, the peripheral cytokine levels in SFTS patients have been associated with disease severity and prognosis. However, as with SFTS viral load, it is difficult to use several cytokines to the early clinical course because of the difficulty surrounding testing in a general laboratory.

This study has several limitations. First, the sample size was relatively small due to the limited availability of Ct values. Increasing the sample size in future studies may provide more robust insight. Second, although we analyzed the correlation between the SFTS viral load and the Ct value, estimating the SFTS viral load within a narrow range based solely on the Ct value remains challenging. Further investigations should consider additional variables. Third, this study focused solely on the initial Ct values; as a result, we were unable to analyze the changes in viral load and Ct values throughout the clinical course of SFTS. Future studies should examine the changes over time. Finally, some patients who were initially predicted to have a fatal prognosis who had initially unstable vital signs, severe thrombocytopenia, multi-organ failure, etc. did not experience a worsening clinical course and were ultimately classified into the non-fatal group. The analysis of the initial SFTS viral load did not reveal a significant difference between the two groups. When considering the calculation of the viral load based on Ct values, particularly in environments where treating fatal patients can be challenging, it is crucial to apply not only this metric but also the patients’ age, condition, and other laboratory results for early severity prediction.

In conclusion, this study contributes to our understanding of SFTS by demonstrating a strong correlation between Ct values and viral load, particularly in fatal cases. These findings offer potential applications for Ct values in predicting disease severity and could aid in more efficient clinical decision making. Nevertheless, further research is needed to validate these findings and explore their broader clinical implications. Overall, our study underscores the importance of continued research in the field of SFTS and the potential use of Ct value as a tool for early prognosis prediction and disease management.

## Figures and Tables

**Figure 1 viruses-16-00700-f001:**
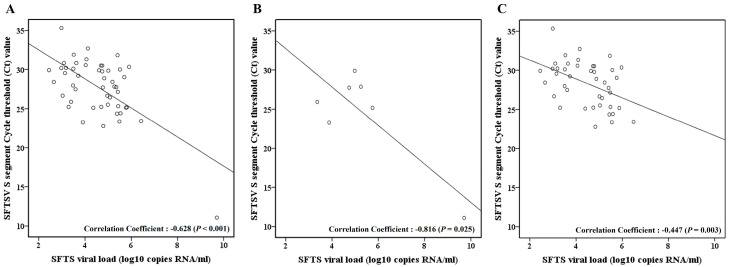
Correlation between severe fever with thrombocytopenia syndrome (SFTS) viral load and short-segment cycle threshold (Ct) value. (**A**) Total SFTS patients (*n* = 49). (**B**) Fatal group (*n* = 7). (**C**) Non-fatal group (*n* = 42).

**Figure 2 viruses-16-00700-f002:**
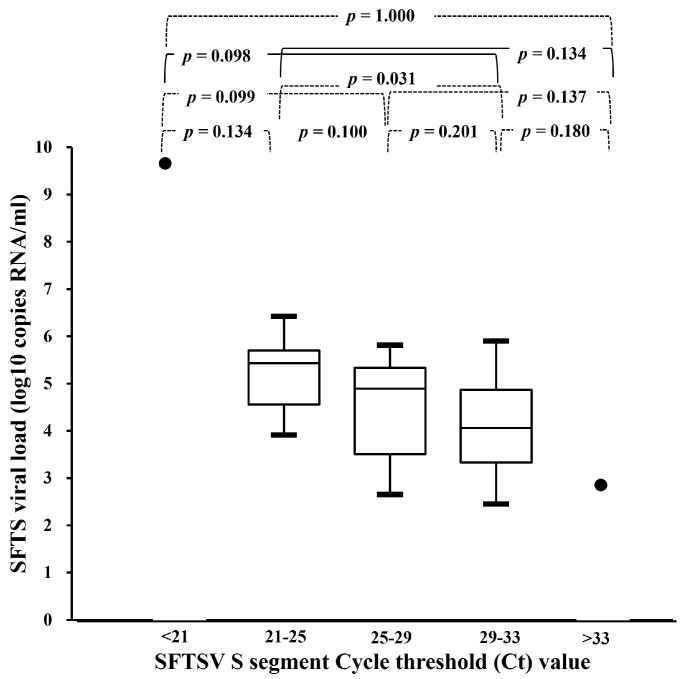
Mean severe fever with thrombocytopenia syndrome (SFTS) viral load (log SFTS viral load) and short-segment cycle threshold (Ct) value by interval. The Ct values were divided into five intervals (<21, 21–25, >25–29, >29–33, and >33), and the mean viral load values were compared for each interval.

**Table 1 viruses-16-00700-t001:** Baseline clinical characteristics of patients with severe fever with thrombocytopenia syndrome between fatal and non-fatal patients.

Variable	All (*n* = 49)	Non-Fatal (*n* = 42)	Fatal (*n* = 7)	*p*-Value
Age, years (mean ± SD)	63.80 (±12.2)	62.8 (±12.3)	69.6 (±10.7)	0.166
Male gender, *n* (%)	29 (59.2)	24 (57.1)	5 (71.4)	0.685
CCI (mean ± SD)	0.5 (±0.8)	0.3 (±0.6)	1.6 (±1.0)	<0.001
Initial log SFTS viral load, log10 copies RNA/mL (mean ± SD)	4.56 (±1.28)	4.42 (±1.05)	5.40 (±2.06)	0.053
Initial SFTS S-segment Ct value (mean ± SD)	27.77 (±3.71)	28.33 (±2.86)	24.38 (±6.25)	0.148
Initial SFTS M-segment Ct value (mean ± SD) ^†^	29.00 (±3.82) ^†^	29.8 (±2.50)	25.23 (±6.45)	0.415
MODS				
Initial	2.5 (±2.6)	2.0 (±1.3)	5.9 ± 5.1	0.092
72 h after diagnosis	4.19 (±3.8)	3.0 (±2.0)	12.3 ± 4.0	0.002
TPE	23 (46.9)	17 (40.5)	5 (71.4)	0.219
Clinical findings upon admission day, *n* (%)				
Fever	42 (85.2)	38 (90.5)	4 (57.1)	0.050
Chill	25 (52.1)	23 (54.8)	2 (33.3)	0.407
Fatigue	20 (41.9)	16 (38.1)	4 (66.7)	0.218
Headache	10 (20.8)	10 (23.8)	0 (0)	0.320
Myalgia	19 (39.6)	18 (42.9)	1 (16.7)	0.381
Dizziness	8 (16.7)	8 (19.0)	0 (0)	0.571
Poor oral intake	24 (50.0)	23 (54.8)	1 (16.7)	0.188
Nausea	14 (29.2)	14 (33.3)	0 (0.0)	0.161
Vomit	9 (18.8)	9 (21.4)	0 (0)	0.578
Abdominal pain	4 (8.3)	4 (9.5)	0 (0)	1.000
Diarrhea	17 (35.4)	15 (35.7)	2 (33.3)	1.000
Hemoptysis	2 (4.2)	1 (2.4)	1 (16.7)	0.237
Dyspnea	2 (4.2)	0 (0)	2 (33.3)	0.013
Laboratory findings (mean ± SD)				
WBC (cells/μL)	2273 (±1850)	1905 (±1260)	4480 (±3149)	0.074
ANC (cells/μL)	1259 (±1080)	1164 (±965)	1910 (±1647)	0.324
Lymphocyte fraction (%)	38.1 (±16.7)	37.5 (±17.6)	41.8 (±9.9)	0.539
Platelet (cells/μL)	85,530 (±38,918)	85,286 (±35,936)	87,000 (±57,350)	0.915
CRP (mg/dL)	0.6 (±1.4)	0.7 (±1.0)	2.2 (±2.5)	0.173
aPTT (sec)	41.4 (±11.3)	39.7 (±6.5)	51.6 (±24.2)	0.245
AST (IU/L)	199 (±344)	103 (±111)	778 (±636)	0.031
ALT (IU/L)	84 (±105)	60 (±54)	235 (±193)	0.053
LDH (IU/L)	1066 (±1352)	717 (±410)	3336 (±2769)	0.068
CPK (IU/L)	1650 (±3577)	1208 (±2356)	3926 (±7068)	0.351

^†^ There are 34 patients with SFTS who underwent M segment testing (non-fatal *n* = 28, fatal *n* = 6). Values are presented as mean ± standard deviation or *n* (%). *p* values are derived from simple *t*-test or Mann–Whitney test for continuous data and Chi-square test for categorical data. CCI, comorbidity index score (calculated by Charlson Comorbidity Index); S-segment, small segment; Ct, cycle threshold; MODS, Multiple Organ Dysfunction Score; TPE, therapeutic plasma exchange; WBC, white blood cell; ANC, absolute neutrophil count; CRP, C-reactive protein; aPTT, activated partial thromboplastin time; AST, aspartate aminotransferase; ALT, alanine aminotransferase; LDH, lactate dehydrogenase; CPK, creatinine phosphokinase.

## Data Availability

The data are not publicly available due to privacy of the patients.

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
