# Peer review of "Correlation between the Cycle Threshold Values in Detection of Severe Fever with Thrombocytopenia Syndrome Virus Using PowerChekTM SFTSV Real-Time PCR Kit and Viral Load: Prognostic Implications"

_viruses, 2024, doi:10.3390/v16050700_

Round 1
Reviewer 1 Report (Previous Reviewer 2)
Comments and Suggestions for Authors
The authors have sufficiently addressed my concerns. No additional comments.
Author Response
I appreciate your review.
Reviewer 2 Report (Previous Reviewer 1)
Comments and Suggestions for Authors
General comments
This manuscript is on the topic of diagnostics for and prediction of the patients with severe fever with thrombocytopenia syndrome virus (SFTS). This reviewer has read this manuscript very carefully and has noticed that there are two main purposes for this study. One is to evaluate the correlation between the Ct values in the SFTSV genome amplification test using a commercially available SFTSV genome detection kit “PowerCheckTM” and SFTS viral load determined with an in-house quantitative real-time PCR for SFTSV genome detection performed in the Department of Microbiology and Immunology of Jeju National University College of Medicine. The test, in which “PowerCheckTM” was used, was carried out in the Jeju Research Institute of Public Health and Environment. The other purpose is to assess the level of Ct values determined using “PowerCheckTM” (PowerCheck Ct values) with the prognosis of the patients with SFTS.
Major comments
1. The term “Ct values” appears in this manuscript are those determined with using PowerCheckTM by the Jeju Research Institute of Public Health and Environment. Please abbreviate the Ct values determined with using PowerCheckTM by the Jeju Research Institute of Public Health and Environment in the entire text. This reviewer wish recommending, for instance, the term be abbreviated as “PowerCheck S-Ct values” or “PowerCheck M-Ct values”.
2. The methods to detect the SFTSV genome with using PowerCheckTM is so critical that the methods should be described more in detail. If the url issued by the company is available, it should be included. The methods how the viral RNA was extracted should also be mentioned; Extracted from total blood; extracted from serum samples? Furthermore, it is written that the viral load was measured in the Department of Microbiology and Immunology of Jeju National University College of Medicine. However, the methods of this quantitative SFTSV RT-PCR are not written. For instance, the sequences of the primers and probe are not described? The readers would be interested in the target viral RNA segment.
3. The difference in the viral genome levels were assessed between non-fatal group and fatal group using the absolute genome copy numbers (Table 1, and text). However, the difference should be statistically evaluated using the viral copy numbers in logarithmic. The line in Table 1 (Initial SFTS viral load, copies RNA/mL (mean+-SD) and the corresponding sentence in the main text can be deleted.
4. There was no statistically significant difference in the viral genome copies between non-fatal group and fatal group with the p-value of 0.053. This reviewer speculates that the difference would become statistically significant, if the number of the cases enrolled would be increased more. The readers may not understand the importance of PowerCheck S-Ct values or PowerCheck M-Ct values logically and rationally in the prediction of the prognosis as the current explanation. It is written that there was a difference in the coefficient values between that assessed PowerCheck S-Ct values and the SFTSV genome load among non-fatal cases (-0.816) and that among the fatal cases (-0.447) (Figure 1). Furthermore, the authors stated in the text that the importance of PowerCheck S-Ct values in the prediction of the prognosis is based on the coefficient values’ difference. The association between PowerCheck S-Ct values and the prediction of the prognosis should logically and rationally be explained. Furthermore, if it is the case, the difference in the coefficient values should be statistically evaluated between the that assessed PowerCheck S-Ct values and the SFTSV genome load among non-fatal cases (-0.816) and that among the fatal cases (-0.447). The authors might misunderstand the reviewer’s comment to the original manuscript by responding with the reply “If you think this is inappropriate, I would appreciate it if you could comment again”. This reviewer hopes the authors understand the content of this comment properly.
5. This reviewer must raise a simple question. Why are neither the contribution of the Jeju Research Institute of Public Health and Environment nor that by the Department of Microbiology and Immunology of Jeju National University College of Medicine described in the authors line-up and the Acknowledgement section.
Specific comments
1. This reviewer recommends to modify the title as follows: “Correlation between the cycle threshold values in detection of severe fever with thrombocytopenia syndrome virus genome using PowerCheckTM real-time PCR kit and viral load: prognostic implication”. Ct values of the M-segment genome detection are not included for analyses in Table 1 except for one line.
2. Line 26, Abstract section: This negative correlation was notably stronger in the fatal group (-0.940, P=0.002) than in the non-fatal group (-0.345, P=0.025). Please make it clear that the difference is assessed statistically. Furthermore, the correlation between the viral load shown in logarithmic and the PowerCheck S-Ct values, but not using the absolute number of viral genome in general.
3. Line 38: The references “2 and 3” seem not appropriate for this description.
4. Line 43: The readers would not understand the description “related to prediction models of viral load”. The sentence should be modified to make the readers understand the content of this sentence clearly.
5. Line 48: Please make it clear that the RT-PCR was carried out using PowerCheckTM here.
6. Line 52-54: This reviewer considers that this sentence should be cited with appropriate references.
7. Line 69: Does the description of “SFTS viral load were assessed on the same day” indicate that both the PowerCheck Ct value assay in the Jeju Research Institute of Public Health and Environment and the SFTS viral load assay performed in the Department of Microbiology and Immunology of Jeju National University College of Medicine on the same day? If it is the case, no corrections are needed.
8. Line 93-95: The methods should be written in the past sense, correcting “is” to “was”.
9. Line 113 and Table 1: If the criteria for offering TPE therapy was present, the criteria why TPE therapy was offered for some patients (not for the others) should be explained.
10. Table 1: The mark at the line “Initial SFTS S-segment Ct value (mean+-SD) at the vertical line position of “All” is not identical to that at the footnote. Careful writing is important.
11. Line 127: “significant statistically” should be “statistically significant”.
12. Line 133-136: Modification or correction is needed as discussed above.
13. Line 164-167, Discussion: The sentence should be with a logical and rational explanation.
14. Line 172-188: Please reconsider whether the discussion on the relationship between viral load and Ct values in terms of SARS-CoV-2 infection (COVID-19) is necessary or not for this manuscript.
15. Line 198, Discussion: “because of difficult to test in general laboratory” needs correction such as “difficulty in testing in general laboratory” and so on.
16. Line 207-209, Discussion: Please specify how and why were “some patients” initially predicted to have a fatal prognosis”. It is better how many were there such patients in the non-fatal group patients.
Comments on the Quality of English LanguageModerate editing of English language required
Author Response
General comments
This manuscript is on the topic of diagnostics for and prediction of the patients with severe fever with thrombocytopenia syndrome virus (SFTS). This reviewer has read this manuscript very carefully and has noticed that there are two main purposes for this study. One is to evaluate the correlation between the Ct values in the SFTSV genome amplification test using a commercially available SFTSV genome detection kit “PowerCheckTM” and SFTS viral load determined with an in-house quantitative real-time PCR for SFTSV genome detection performed in the Department of Microbiology and Immunology of Jeju National University College of Medicine. The test, in which “PowerCheckTM” was used, was carried out in the Jeju Research Institute of Public Health and Environment. The other purpose is to assess the level of Ct values determined using “PowerCheckTM” (PowerCheck Ct values) with the prognosis of the patients with SFTS.
Author response: Thank you for your detailed review and comments. I am concerned whether the author response can satisfy about all your comments, but I tried to answer as best as I could.
Major comments
1. The term “Ct values” appears in this manuscript are those determined with using PowerCheckTMby the Jeju Research Institute of Public Health and Environment. Please abbreviate the Ct values determined with using PowerCheckTM by the Jeju Research Institute of Public Health and Environment in the entire text. This reviewer wish recommending, for instance, the term be abbreviated as “PowerCheck S-Ct values” or “PowerCheck M-Ct values”.
Author response: Thank you for your comment. In the methods section, I modified the sentence as follows: The Ct values of the S and M segment gene of the SFTSV RNA were evaluated using the SFTSV real-time RT-PCR Kit (PowerChek; Kogene Biotech, Seoul, Republic of Korea) at the Jeju Research Institute of Public Health and Environment (Powercheck S-Ct value and M-Ct value, hereafter referred to S-segment and M-segment Ct value in the text).
2. The methods to detect the SFTSV genome with using PowerCheckTM is so critical that the methods should be described more in detail. If the url issued by the company is available, it should be included. The methods how the viral RNA was extracted should also be mentioned; Extracted from total blood; extracted from serum samples? Furthermore, it is written that the viral load was measured in the Department of Microbiology and Immunology of Jeju National University College of Medicine. However, the methods of this quantitative SFTSV RT-PCR are not written. For instance, the sequences of the primers and probe are not described? The readers would be interested in the target viral RNA segment.
Author response: Thank you for your comment. Please understand that we cannot find the published URL related to genome detection by the Powercheck company, so we cannot attach it or explain it in detail. And regarding the method of extracting viral RNA, I think the process and kit used were sufficiently explained in the method section. It was already mentioned that the patient's first acute-phase serum was used for virus RNA detection, and the currently commercially available BigDye Terminator Cycle Sequencing kit was used for the sequence. I’m not sure that it necessary to mention more detailed testing methods in the brief report, as the focus of this study is not on the new methodology for detecting SFTS virus or Ct value.
3. The difference in the viral genome levels were assessed between non-fatal group and fatal group using the absolute genome copy numbers (Table 1, and text). However, the difference should be statistically evaluated using the viral copy numbers in logarithmic. The line in Table 1 (Initial SFTS viral load, copies RNA/mL (mean+-SD) and the corresponding sentence in the main text can be deleted.
Author response: Thank you for your comment. I delated the context about absolute genome copy numbers in Table and text.
4. There was no statistically significant difference in the viral genome copies between non-fatal group and fatal group with the p-value of 0.053. This reviewer speculates that the difference would become statistically significant, if the number of the cases enrolled would be increased more. The readers may not understand the importance of PowerCheck S-Ct values or PowerCheck M-Ct values logically and rationally in the prediction of the prognosis as the current explanation. It is written that there was a difference in the coefficient values between that assessed PowerCheck S-Ct values and the SFTSV genome load among non-fatal cases (-0.816) and that among the fatal cases (-0.447) (Figure 1). Furthermore, the authors stated in the text that the importance of PowerCheck S-Ct values in the prediction of the prognosis is based on the coefficient values’ difference. The association between PowerCheck S-Ct values and the prediction of the prognosis should logically and rationally be explained. Furthermore, if it is the case, the difference in the coefficient values should be statistically evaluated between the that assessed PowerCheck S-Ct values and the SFTSV genome load among non-fatal cases (-0.816) and that among the fatal cases (-0.447). The authors might misunderstand the reviewer’s comment to the original manuscript by responding with the reply “If you think this is inappropriate, I would appreciate it if you could comment again”. This reviewer hopes the authors understand the content of this comment properly.
Author response: Thank you for your comment. I understand your intention. What I want to present, this analysis that the more negative the size of the correlation coefficient, the stronger the negative correlation, and the more negative correlation coefficient in the fatal group. However, I’m not sure that it is meaningful to additionally verify the statistical significance of the negative correlation coefficient between the fatal and non-fatal groups.
5. This reviewer must raise a simple question. Why are neither the contribution of the Jeju Research Institute of Public Health and Environment nor that by the Department of Microbiology and Immunology of Jeju National University College of Medicine described in the authors line-up and the Acknowledgement section.
Author response: Thank you for your comment. I made the mistake of overlooking this importance and corrected it right away.
Specific comments
1. This reviewer recommends to modify the title as follows: “Correlation between the cycle threshold values in detection of severe fever with thrombocytopenia syndrome virus genome using PowerCheckTMreal-time PCR kit and viral load: prognostic implication”. Ct values of the M-segment genome detection are not included for analyses in Table 1 except for one line.
Author response: Thank you for your comment. I modified the title as your recommend.
2. Line 26, Abstract section: This negative correlation was notably stronger in the fatal group (-0.940, P=0.002) than in the non-fatal group (-0.345, P=0.025). Please make it clear that the difference is assessed statistically. Furthermore, the correlation between the viral load shown in logarithmic and the PowerCheck S-Ct values, but not using the absolute number of viral genome in general.
Author response: The more negative the size of the correlation coefficient, the stronger the negative correlation, and the more negative the correlation coefficient in the fatal group. The sentence has been modified to be more clearly.
3. Line 38: The references “2 and 3” seem not appropriate for this description.
Author response: The reference is corrected more appropriately.
4. Line 43: The readers would not understand the description “related to prediction models of viral load”. The sentence should be modified to make the readers understand the content of this sentence clearly.
Author response: Thank you for your comment. The description is modified.
5. Line 48: Please make it clear that the RT-PCR was carried out using PowerCheckTMhere.
Author response: The context is added.
6. Line 52-54: This reviewer considers that this sentence should be cited with appropriate references.
Author response: This sentence is about the hypothesis in this study.
7. Line 69: Does the description of “SFTS viral load were assessed on the same day” indicate that both the PowerCheck Ct value assay in the Jeju Research Institute of Public Health and Environment and the SFTS viral load assay performed in the Department of Microbiology and Immunology of Jeju National University College of Medicine on the same day? If it is the case, no corrections are needed.
Author response: The sentence was revised “Patients whose Ct values of the S and M segments of the SFTSV RNA and SFTS viral load were assessed on the using same day of serum sample”
8. Line 93-95: The methods should be written in the past sense, correcting “is” to “was”.
Author response: It was corrected.
9. Line 113 and Table 1: If the criteria for offering TPE therapy was present, the criteria why TPE therapy was offered for some patients (not for the others) should be explained.
Author response: In the case of TPE therapy, it is difficult to explain about the criteria because it tended to be performed empirically in patients who showed a clinical worsening trend rather than based on clear standards.
10. Table 1: The mark at the line “Initial SFTS S-segment Ct value (mean+-SD) at the vertical line position of “All” is not identical to that at the footnote. Careful writing is important.
Author response: I think you mentioned about “SFTS M-segment”. I corrected to the same marker in table and footnote.
11. Line 127: “significant statistically” should be “statistically significant”.
Author response: It was corrected.
12. Line 133-136: Modification or correction is needed as discussed above.
Author response: What this analysis means that the more negative the size of the correlation coefficient, the stronger the negative correlation, and the more negative correlation coefficient in the fatal group. I don’t think that further modifications to this content are necessary.
13. Line 164-167, Discussion: The sentence should be with a logical and rational explanation.
Author response: Thank you for your comment. This sentence is a summary of the analysis in the result, but I do not think it is illogical and unreasonable as the result has already been mentioned before
14. Line 172-188: Please reconsider whether the discussion on the relationship between viral load and Ct values in terms of SARS-CoV-2 infection (COVID-19) is necessary or not for this manuscript.
Author response: Among other viral infections a recent study that analyzed the relationship between the viral load and Ct value of COVID-19 showed similar results (linear correlation) of our study, we think it would be helpful for discussion.
15. Line 198, Discussion: “because of difficult to test in general laboratory” needs correction such as “difficulty in testing in general laboratory” and so on.
Author response: It was corrected as your recommendation.
16. Line 207-209, Discussion: Please specify how and why were “some patients” initially predicted to have a fatal prognosis”. It is better how many were there such patients in the non-fatal group patients.
Author response: Rather than predicting a fatal prognosis based on clear criteria, it was judged that patients with initially unstable vital signs, very severe thrombocytopenia or multi-organ failure were likely to have a poor prognosis in the early stages, but depending on the patient's age, underlying disease, and level of conservative management, some patients finally survived.

This manuscript is a resubmission of an earlier submission. The following is a list of the peer review reports and author responses from that submission.
Round 1
Reviewer 1 Report
Comments and Suggestions for Authors
General comments
The authors studied the relationship between the Ct values of the serum samples collected in the early phase of the severe fever with thrombocytopenia syndrome (SFTS) patients determined by a commercially available SFTS virus genome detection kit (PowerCheck Kogene Biotech, Seoul, ROK) and the SFTS viral load determined in the Department of Microbiology and Immunology, Jeju National University. The patients enrolled in this study were those who were diagnosed as having SFTS in the University between March 2016 and December 2022. Sixty-seven patients might had been diagnosed as having SFTS during the study period and 49 of them were enrolled by the inclusion criteria defined by the authors. The patients enrolled were divided into two groups of fatal (7 patients) and non-fatal (42 patients) groups. The authors also studied the relationship between the Ct values and the prognosis.
This reviewer considers that this study might be of value in the clinical setting for the treatment of patients with SFTS and scientifically.
However, the study purpose, design and interpretation of the results are premature or wrong for publication as the current form.
1) The Ct values, which appears in the title and the entire text might be specific to those determined with the commercially available SFTS virus genome detection kit (PowerCheck Kogene Biotech, Seoul, ROK), are not of general, and cannot be generalized. Therefore, the kit brand name should be included in the title.
2) The commercially available SFTS virus genome detection kit (PowerCheck Kogene Biotech, Seoul, ROK) should be explained in detail in terms of the mechanism of genome detection, sequences of the primers and probes, and amplification conditions.
3) It should be made clear whether the viral load can be calculated with the kit, PowerCheck or not. In general, viral load can be calculated if the Ct values can be determined in a real-time manner.
4) It is written that the viral load was determined in the in the Department of Microbiology and Immunology, Jeju National University, but the methods are not described in the Materials and Methods section, although the method is very critical and important for this study.
5) M-segment-associated Ct value is included as one of the topics in the study. However, there are no descriptions on the M-segment-associated Ct value. Only the analyses were focused on S-segment-associated Ct values.
6) The authors evaluated the efficacy of the Ct values in association of viral genome load shown in absolute numbers and the logarithm of the absolute number. This reviewer considers that the viral load of SFTSV in the blood should be shown in a logarithmic manner.
7) It should clearly be mentioned that this study was conducted in a retrospective or prospective manner in the Materials and Methods section, although it is written in the last part of the manuscript.
8) Line 117-120: The authors might analyze the statistical difference of the quality of association between the evaluation of SFTS viral genome load shown in absolute number with the Ct value and that of SFTS viral genome load shown in a logarithmic manner with the Ct value by comparing the correlation coefficients and p-values. However, this reviewer considers that the analytical method for evaluating the difference is not appropriate for the statistical evaluation.
9) The authors stated that there was a statistical difference between the SFTS viral load shown logarithmically between the non-fatal cases and fatal cases with the p-value of 0.05. If the definition of the p-value for the statistical difference was less than 0.05, it indicates that the SFTS viral load shown logarithmically between the non-fatal cases and fatal cases should not be statistically different.
Specific comments
1) As explained above, the title needs correction.
2) Line 24-25: The statistical analyses for the difference between the negative correlation between fatal group and non-fatal group should be based on appropriate methods.
3) Line 36: The sentence “SFTS has no specific treatment” does not make sense, because SFTS is just the disease name. It is evident that there are no specific treatments for SFTS at this moment or that we have no specific treatment for patients with SFTS.
4) Line 48: Does the guideline issued from Korean CDC required us to diagnose someone has having SFTS by detection of both S- and M-segment RNA? Is it true?
5) Line 56: “longitudinal observational” should be “longitudinal and observational”.
6) Line 63-64: The descriptions of CCI and MODS should be accompanied with the appropriate references.
7) Line 70: The sentence “A Ct value equal to or less than 38 is reported as positive for SFTS” does not make sense. Samples with the Ct value of equal to or less 38 was reported as positive for SFTS viral-genome detection.
8) Line 77: The description “between study population” does not make sense. “Between” should be “among”? Please specify the targets for comparison between “A” and “B”, if the word “between” would be used here.
9) Line 91: The data of (1.6+-1.0 vs. 0.3+-0.6, P<0.001) in Line 91 is written in Table 1 as well. This description should be deleted to avoid the dual descriptions as much as possible. This kind of correction is needed in the same phrase.
10) Line 102, Table 1: There is no significant difference in the initial SFTS S-segment Ct value with the p-value of 0.148. Does this data suggest or indicate that the Ct value can not suggest or indicate the prognosis in SFTS patients? Interpretation of this result is important. Careful interpretation of this data is needed.
11) Line 102, Table 1: Is the P-value of 0.05 at the column of Initial log SFTS viral load indicate statistically different or not statistically different? Careful interpretation with a clear definition is needed.
12) Line 122, Figure 1B: This reviewer can see only 6 dots in Fig 1B. Should the dot numbers appear in the figure be 7?
13) Line 122, Figure 1: These 3 figures can be combined, making the figure one.
14) Line 136, Figure 2. The Y-axis of Figure 2A is based on the viral load shown as the absolute number. This reviewer considers that this figure should not so be important. Figure 2B is enough. Furthermore, In Fig. 2A, Table and Figure is included. The Table or the Figure can be deleted.
15) Line 147: The term “Cutoff value” appears here, but this reviewer could not find the description of the “Cutoff value”. It should be made clear what the term “Cutoff” indicates.
16) Line 170: Please confirm that reference 12 is “your” study. Are all the authors in this study identical to those of the reference 12?
17) Line 191: According to the discussion on the association between the outcome and TPE treatment, the authors may consider that TPE is effective as the specific treatment for patients with SFTS. However, this reviewer considers that there is no evidence on the effectiveness of TPE treatment for patients with SFTS. Careful discussion is needed.
Comments on the Quality of English LanguageSome sentences needs corrections as the scientifically correct descriptions.
Reviewer 2 Report
Comments and Suggestions for Authors
Overall the paper is well-written and minimal revisions are suggested. The two concerns which are detailed in the comments on the attached manuscript are listed below.
1. Separate the two ideas in the discussion to accurately reflect the supporting references.
2. In the discussion section, the authors need additional context for the impact of the day post-infection on Ct values / virus titers.

Reviewer 3 Report
Comments and Suggestions for Authors
The manuscript titled " Correlation between the Cycle Threshold Values of Medium- and Small-segment Severe Fever with Thrombocytopenia Syndrome Virus and Viral Load: Prognostic Implications" by Kim M. et al. discusses the evaluation of Ct values of SFTS viral RNA (S or M RNA) from patients, aiming to establish a Ct value cutoff for estimating initial viral load and predicting prognosis early. This study characterized patient samples based on clinical outcomes (non-fatal vs. fatal), age (≥18y), availability of blood samples, or available viral load and Ct values on the same day. The authors demonstrated a negative correlation between SFTS viral load and Ct values (i.e., low Ct values indicate high viral loads) for S and M segment RNA. The Ct values for S and M RNA are part of the qPCR testing for "SFTS viral loads (copies RNA/ml)." Thus, "SFTS viral loads" reflect the normalization of sample input among samples. Nonetheless, direct evaluation of Ct values without normalization of sample input may decrease the accuracy of viral load evaluation, potentially not improving clinical laboratory practice by itself.
Additional comments:
· The method for measuring "SFTS viral loads (copies RNA/ml)" should be clearly described in the Materials and Methods section.
· Table 1: The discrepancy between "Initial log SFTS viral loads" and "Initial SFTS viral loads" (e.g., 4.42 vs. 5.40 in log10 corresponding to 184341 vs. 724100062) needs clarification.
· Line 35: The classification of SFTS virus should be described.
· Line 64: The authors should clarify whether all patients (n = 49) in this study exhibited clinical signs of the disease.
· Table 1: Criteria for inclusion in "clinical findings" should be explained. Authors should specify whether patients were included only if they exhibited relevant clinical signs of the disease upon admission to the hospital (or any day during admission).
· Further details of "fatal" patients should be provided, including survival duration or major differences in clinical signs compared to non-fatal patients.
· The apparent lack of reduction in platelet counts among listed SFTS patients should be mentioned in the text.
· Authors should provide a description of the approval status concerning the retrospective analysis of patients' data.